# Concept of Sustainable Demolition Process for Brickwork Buildings with Expanded Polystyrene Foam Insulation Using Mealworms of *Tenebrio molitor*

**DOI:** 10.3390/ma15217516

**Published:** 2022-10-26

**Authors:** Sebastian W. Przemieniecki, Jacek Katzer, Agnieszka Kosewska, Olga Kosewska, Paweł Sowiński, Paulina Żeliszewska, Barbara Kalisz

**Affiliations:** 1Faculty of Agriculture and Forestry, University of Warmia and Mazury in Olsztyn, 10-719 Olsztyn, Poland; 2Faculty of Geoengineering, University of Warmia and Mazury in Olsztyn, 10-719 Olsztyn, Poland; 3Jerzy Haber Institute of Catalysis and Surface Chemistry Polish Academy of Sciences, 30-239 Krakow, Poland

**Keywords:** construction, waste, demolition, *Tenebrio molitor*, brick and mortar

## Abstract

The traditional demolition process for brickwork buildings results in a significant volume of mixed debris. The debris consists of ceramic bricks (and other wall elements), mortar, thermal insulation (usually expanded polystyrene or rockwool), smaller steel elements, pieces of wood, and glass. Such mixed debris is difficult to recycle. Separating thermal insulation that is “glued” by cement mortar to brickwork is probably the most difficult and time-consuming task in processing mixed debris. This task can be performed in a very different and fully “automatized” manner using *Tenebrio molitor* mealworms. The mealworms remove expanded polystyrene from brickwork surfaces and transform it into frass. In the paper, a research program aiming to prove the concept of using the mealworms of *Tenebrio molitor* for processing mixed debris is presented. The tests were conducted using two models of a three-layered brickwork wall, which is very common in Europe. The proposed approached was successful. Both types of used expanded polystyrene foam (EPS) were fully removed from multilayer wall specimens. The possibilities and limitations of the proposed processing method were discussed and analyzed. The conducted research proved that it is feasible to clean brickwork debris from the EPS using *Tenebrio molitor* mealworms. Differences in the speed of cleaning process regarding the type of EPS were noted. More research is needed to scale the process, and to find the best method for using frass. By using *Tenebrio molitor* mealworms, one can make the demolition process much cleaner.

## 1. Introduction

The construction sector is one of the largest in the world and, globally, generates huge amounts of construction and demolition waste (CWD). In Europe, the majority of the 3 billion tons of CWD generated every year [1,2] is associated with traditional brick and mortar structures. There is increasing pressure to minimize the amounts of CWD created, especially in the European Union. There are many programs and directives regulating this issue, including the Waste Framework Directive 2008/98/EC and end-of-life management [3], as well as Zero Waste [4] based on the circular economy [5]. Usually, the demolition process for a brick-and-mortar building is a very basic one. Walls are easily destroyed using ordinary diggers (see Figure 1). The debris produced during such a demolition process is a mixture of red ceramics (bricks, hollow wall elements, etc.), cement mortar, thermal insulation, timber, steel, and glass. A lack of segregation of materials during the demolition process significantly limits the possible recycling of such debris.

Red ceramics (even those significantly contaminated by cement mortar) are reasonably easy to transform into recycled lightweight aggregate that can be used for the production of new concrete. This process has been thoroughly described by multiple research teams [6,7]. Probably the most interesting approach to harness crushed red ceramic elements as waste aggregate is associated with utilization of an internal curing process [8]. It is also possible to recycle other waste materials, such as mortar, concrete, cement, or lime plaster, and use them as micro-fillers or powders that can partially substitute cement [9]. The most problematic from the recycling perspective is thermal insulation (usually in the form of expanded polystyrene foam (EPS)) glued to bricks by ordinary or highly specialized cement mortar. During the process of traditional demolition, pieces of such a multilayer wall contaminate the debris and are very difficult to handle. Only manual processing guarantees the effective removal of EPS from bricks and cement mortar. On the other hand, manual processing is so expensive that its feasibility is doubtful (especially on a full industrial scale). In the authors’ opinion, this situation is unacceptable and should be resolved in an innovative and cross-disciplinary way. The authors propose the harnessing of mealworms (*Tenebrio molitor*) [10,11] to remove EPS from bricks and mortar. Subsequently, clean bricks with the remains of mortar will be transformed into waste aggregate or fillers for concrete production. If the proof of concept is successful, the possible utilization of the proposed approach for recycling other types of plastics is also possible. 

The aim of this study was to evaluate the potential for EPS utilization by mealworms, from the civil engineering perspective. Characteristics and possible usability for the future application of the obtained frass were also in the focus of the current study.

## 2. Materials and Methods

Mealworms are the larval form of the beetle *Tenebrio molitor*. This is a species of darkling beetle representing holometabolic insects [12]. *Tenebrio molitor* goes through four life stages: egg, larva, pupa, and adult. An average larva usually measures approximately 25 mm. An adult insect usually measures about 15 mm in length. Both the larval form of the mealworm and the adult insect are presented in Figure 2.

The mealworm larvae are distinguished from other larvae of holometabolic insects by their voracity, resistance to unfavorable environmental conditions, and rapid biomass growth [13]. *Tenebrio molitor* is characterized by the ability to digest foods that are not naturally present in the environment, including polyethylene, polystyrene, and processed cellulose, as confirmed by many research studies [14,15]. Even plastics, which are durable and resistant to mechanical damage and biodegradation, can be used by these insects as food. Mealworms can chew plastic due to their strong mandibles [16,17]. Their gut contains symbiotic microbiota which support the digestive process, playing a very important role in the digestion of hard-to-degrade plastics [18,19].

Two types of EPS were used during the research program: ordinary white EPS and grey EPS with the addition of graphite. Ordinary white EPS is of general construction use for insulating walls, floors, etc. EPS with the addition of graphite is characterized by higher thermal properties in comparison to ordinary white EPS. Therefore, it is mainly used for the erection of energy efficient buildings. The basic properties of both EPS types used are summarized in Table 1. EPS prisms were glued to the ceramic bricks using ordinary cement mortar characterized by a water/cement (*w*/*c*) ratio of 0.5. The mortar was prepared using natural post-glacial sand, thoroughly described in previous publications [20], and tap water. Ordinary red ceramic bricks were used to model the external and internal layers of a brickwork wall. The bricks were characterized by a compressive strength of 10 MPa and an apparent density of 1800 kg/m^3^. 

Both EPS specimens were of the same size. They were glued to a ceramic brick from both sides using mortar. In this way, the most common multilayer (ceramic brick, cement mortar, thermal insulation, cement mortar, and ceramic brick) external wall was modelled. Specimens placed in glass containers just before the tests and after adding mealworms are presented in Figure 3 and Figure 4, respectively.

*Tenebrio molitor* was added to both glass containers, with the same number of 1000 individuals (±3%). The experiment was conducted under lab conditions (temp. +20 °C, r.h. +45%). The glass containers were exposed to natural daylight cycles. The specimens were left for 14 days in the glass containers without any interruption from the research team. Specimens at the end of the experiment are presented in Figure 5.

After the experiment, *Tenebrio molitor* and truss were removed separately from the glass containers. The size distribution of the truss particles was measured using a Beckman Coulter (Indianapolis, IN, USA) LS 13 320 particle size analyzer. The low-angle forward light scattering with additional polarization intensity differential scattering (PIDS) technology was used to measure the particle sizes. Vertical and horizontal polarized light at six different angles using three additional wavelengths were utilized for the measurements. The full implementation of both Fraunhofer and Mie theories [21] was applied. This state-of-the-art laser-based technology enables the analysis of particles without missing either the largest or the smallest particles in a sample. The profiles of chemical changes of synthetic polymers during digestion by larvae were identified using Fourier transform infrared spectroscopy (FTIR). For the detection of IR absorption, raw samples of frass were placed in a FTIR detector (Thermo Scientific™ Nicolet™ iS20 FTIR Spectrometer produced by ThermoFisher Scientific, Waltham, MA USA). Pieces of both types of EPS were used as a control for the respective frass samples. The results were collected and analyzed using OMNIC v9 software. The relationships between observations were determined using principal component analysis (PCA) [22] based on the Pearson correlation-similarity matrix. Agglomerative hierarchical clustering (AHC) was obtained based on the Bray and Curtis dissimilarity matrix, and dendrogram construction was performed using Ward’s method. The results were processed statistically and interpreted graphically in XLSTAT [23].

## 3. Results

The main objective of the proof-of-concept research program was achieved. A layer of EPS glued from both sides by cement mortar to ceramic bricks was removed by the *Tenebrio molitor* larvae. In Figure 6, surfaces of the cement mortar that were in contact with EPS are presented. As can be observed in Figure 5 and Figure 6, (see also Appendix A: Timelapse) the process was quicker in the case of the grey EPS (the mortar surface was completely cleaned). In the case of the white EPS, some residue was still present on the mortar surface. Keeping the specimen with white EPS slightly longer in the presence of the *Tenebrio molitor* larvae would have also resulted in the full clearing of the mortar surface. Bricks with cement mortar with fully removed grey EPS are ready for transformation into waste aggregate and fillers for concrete production.

In Figure 7, microscopic images of frass obtained after the consumption of both EPS are presented. Apart from the pure frass, the presence of unprocessed EPS particles and insect body residues was observed. The presence of insect body residues should be associated with cannibalism of the larvae. The particle size distribution of frass was tested in the range from 0.040 µm to 2000 µm. The distributions of the particle diameters are presented in Figure 8. Other statistical characteristics of the frass and larva population are presented in Table 2.

After the granulometric tests of frass, a FITR analysis was performed. The results for white and grey EPS were characterized by 16 and 20 picks, respectively (the detailed results of these tests are presented in Appendix B). The pH values for both frass types were similar. The white EPS frass was characterized by a value of 7.95, and the grey EPS frass by a value of 7.74. The chemical composition of frass is presented in Table 3. Six chemical compounds (C, N, P_2_O_5_, Ca, and Mg) constitute 99.4% of the frass. 

The results presented in Table 2 and Table 3 were used for the PCA analysis, which showed that the frass parameters obtained after the processing of both white and grey EPS were very similar (*r^2^* = 0.996). The created PCA biplot (see Figure 9) composed of the components F1 and F2 obtained 100% of the initial data variance. Based on the properties presented in Table 2 and Table 3, and the PCA results, it was observed that the white EPS processed by the larvae had a stronger effect on larvae mortality than the grey EPS. It was also characterized by a larger diameter of the produced frass. On the other hand, frass from grey EPS was characterized by a higher content of phosphorus, a larger mass of larvae after two weeks of experiment, and a larger number of chemical modifications of frass, in comparison to the unprocessed EPS. The contents of other elements, pH, and mass of the obtained frass were similar, regardless of the type of consumed EPS.

In Figure 10, a dendrogram of agglomerative hierarchical clustering (AHC) is presented. This shows the similar distributions of the results of individual variables. The first clade (left side) was characterized by a low variability and consisted of such variables as: mortality, the mass and size of frass, and pH, as well as elements (Na, C, N, Mg, and Pb). It was created for the elements with the lowest share in relation to all the results. The second clade (right side) was characterized by larger differences between the variables, and consisted of: the mass of the larvae, the number of chemical modifications of the frass, and the elements (Ni, Ca, Al, Co, Cr, K, Fe, Cd, Mn, P, and Zn). It was divided into two groups (red). The left branch consisted of pH and elements with an average proportion (Mn, Zn, Fe, Na, Al, and Mg), with the proportions of Al and Fe concentrations being larger for white EPS. The second group was split into two subgroups, the first one consisting of mortality and size, and the second one consisting of frass mass, Ca, modification, and K.

## 4. Discussion

The conducted research program proved that it is feasible to clean brickwork debris from the EPS using *Tenebrio molitor* larvae. Cleaned brickwork is reasonably easy to further recycle (using ordinary methods) and transform into aggregate for concrete or into fume for cement. Frass as a waste material from the processing of EPS constitutes only a fraction of the initial volume of EPS (less than 10%). There are no new harmful or toxic organic residues in the frass (in comparison to the chemical composition of the EPS, which is an approved building material). Therefore, the proposed biodegradation process of EPS should be considered as safe. Future research programs should be focused on its possible applications, taking into account its chemical composition and properties. In the construction industry, it could be used as a thermal insulating material or the equivalent of some types of lightweight aggregates. It should be remembered that the mealworm is an accepted food source for animals [24]. Therefore, the impact of contamination of insect feed with plastic may indirectly affect other organisms (including humans). Frass is used in agriculture as a fertilizer [25], but the scale of this application is small. 

The degree of larvae activity and the consumption rate of both types of EPS was similar, and both objects were almost completely depleted of EPS after 14 days of exposure. A slightly weaker dynamic of white EPS consumption was observed, which is caused by its larger material density. Some EPS residues were present, and the frass mass was smaller. Taking into consideration the results presented in Table 2 and the AHC analysis, one can conclude that the mortality is inversely correlated with the weight of the larvae and the number of chemical modifications of frass. Moreover, in the variant with white EPS, there are higher volumes of C and N present, suggesting a higher content of dead larvae in this variant. Grey EPS (with graphite) probably contains a higher content of elements necessary for insects to carry out their life processes. The differences between both options are marginal, and both the dynamics of EPS consumption and the welfare of the insects are satisfactory and comparable in both variants. 

Polystyrene is a particularly onerous type of plastic. Approximately 35 million tons of it are produced annually. The production of polystyrene continues to increase by about 5% per year [26]. A majority of the polystyrene is created as expanded polystyrene foam insulation and is consumed by the construction industry. Over the years, the thermal requirements for housing (e.g., in Europe) have been drastically increased to save the amount of energy used in heating and air conditioning. EPS is used in significantly growing volumes to meet thermal requirements. When the lifecycle of the currently erected building comes to an end, the CDW achieved during the process of demolition will be much more contaminated by EPS than it is currently. The development of a sustainable (associated with low energy consumption) and preferably autonomous method of separating EPS from brickwork debris is essential for the future of the construction industry. The proposed method of harnessing *Tenebrio molitor* ‘larvae fulfils these requirements. The executed tests proved the feasibility of the method on a laboratory scale. More research is needed to scale the process and to find the best method for using frass. The construction of a dedicated chamber for the process of EPS “removal” is inevitable. In this close and controlled environment, the recuperation of heat produced by the mealworms consuming the EPS may be possible. The process of EPS removal using mealworms is also fully autonomous, contributing to savings in labor and energy usage. 

## 5. Conclusions

The following conclusions can be drawn from the conducted tests:It is feasible to clean brickwork debris from EPS using *Tenebrio molitor* larvae.The process of cleaning brickwork using *Tenebrio molitor* larvae is faster in the case of grey EPS, in comparison to white EPS.There are no new harmful or toxic organic residues in the frass. Therefore, the proposed biodegradation process for EPS should be considered as safe.More research is needed to scale the process and to find the best method for using frass.

## Figures and Tables

**Figure 1 materials-15-07516-f001:**
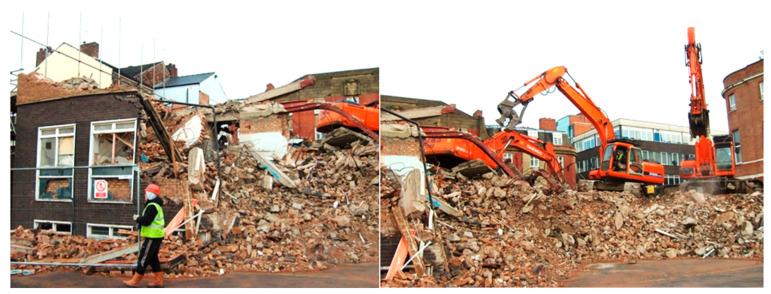
Exemplary European demolition site (photo by J. Katzer).

**Figure 2 materials-15-07516-f002:**
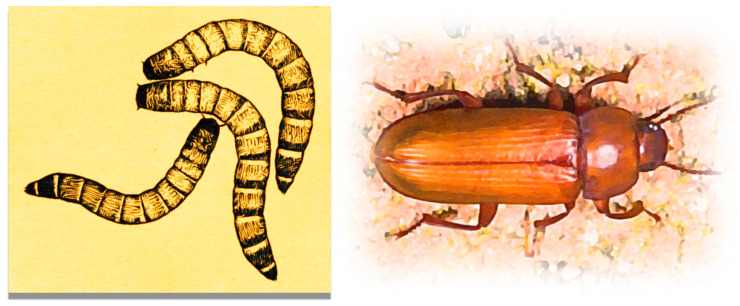
The larval form of the mealworm (**left**) and the adult insect (**right**).

**Figure 3 materials-15-07516-f003:**
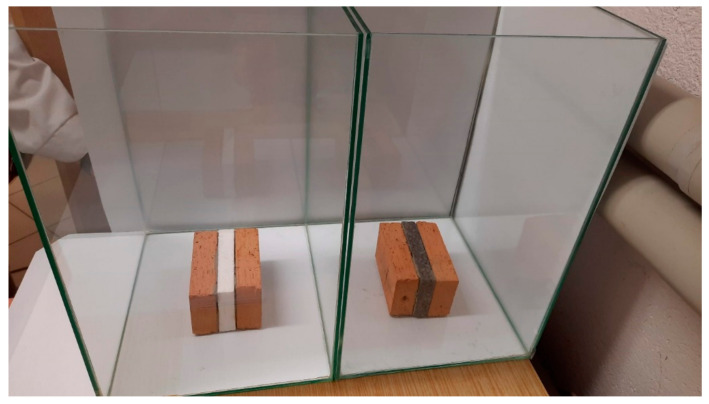
Specimens placed in glass containers.

**Figure 4 materials-15-07516-f004:**
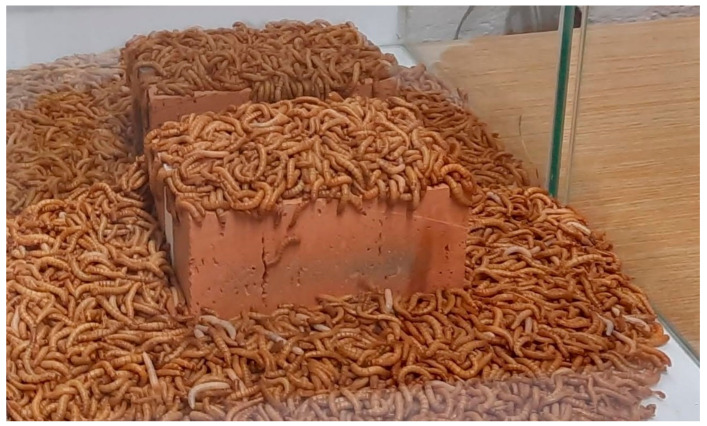
Specimens just after placing *Tenebrio molitor* in glass containers.

**Figure 5 materials-15-07516-f005:**
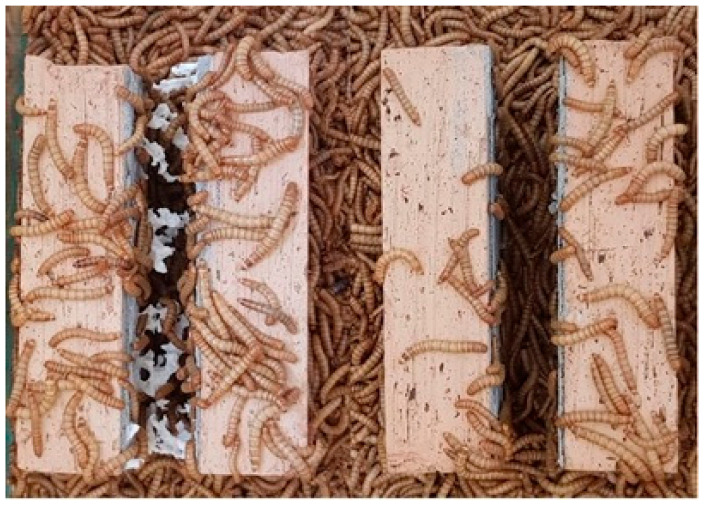
Specimens at the end of the experiment (left—with white EPS, right—with grey EPS).

**Figure 6 materials-15-07516-f006:**
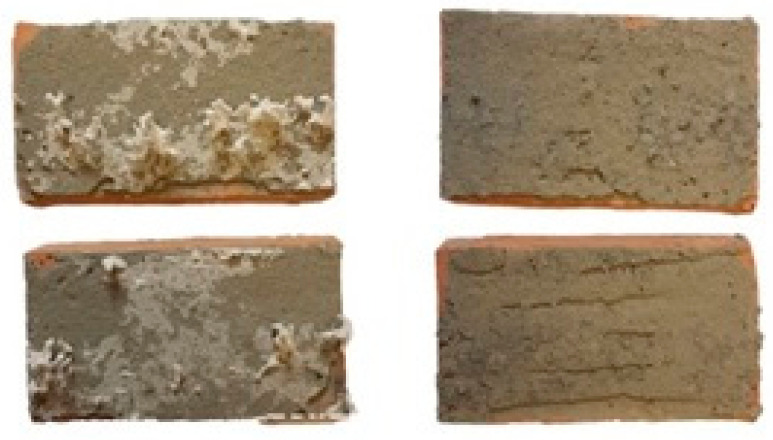
Surfaces of cement mortar glued to EPS (left—white, right—grey).

**Figure 7 materials-15-07516-f007:**
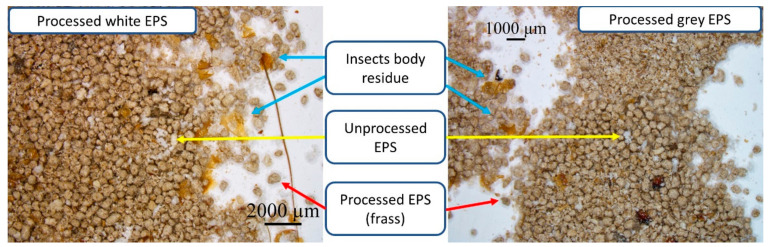
Microscopic images of frass obtained after the consumption of white and grey EPS by *Tenebrio molitor* larvae.

**Figure 8 materials-15-07516-f008:**
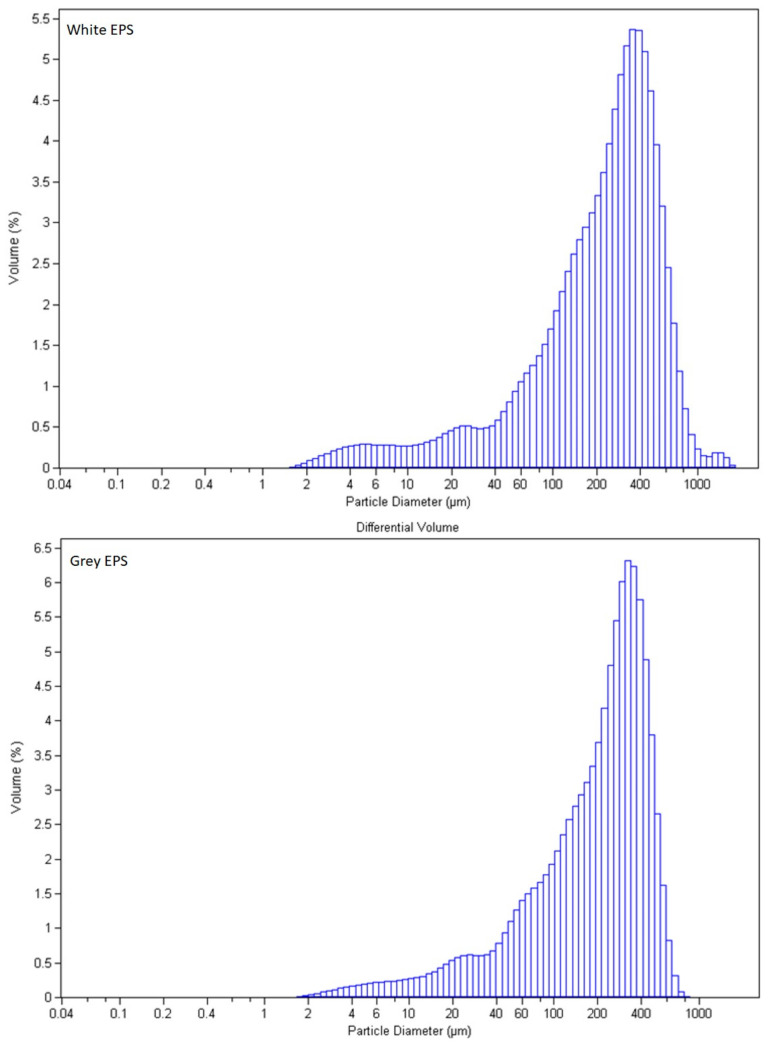
Particle size distribution of white and grey EPS frass.

**Figure 9 materials-15-07516-f009:**
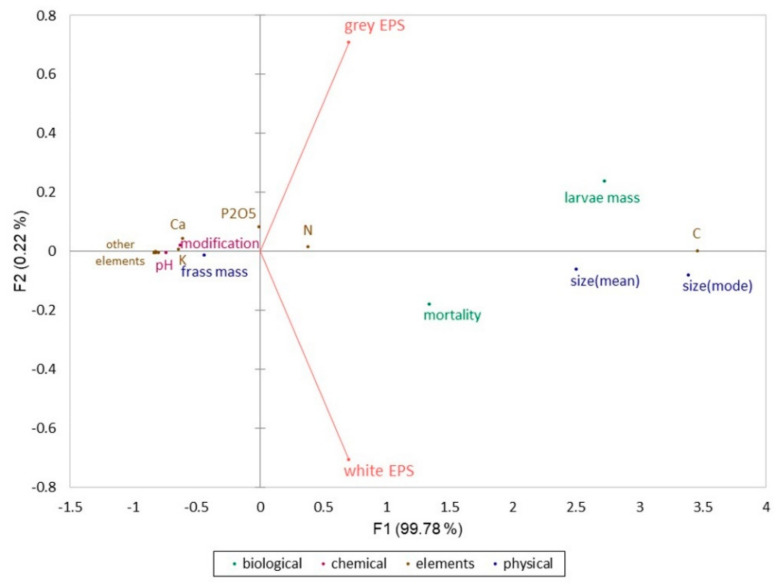
PCA biplot showing relationship between frass obtained from both types of EPS.

**Figure 10 materials-15-07516-f010:**
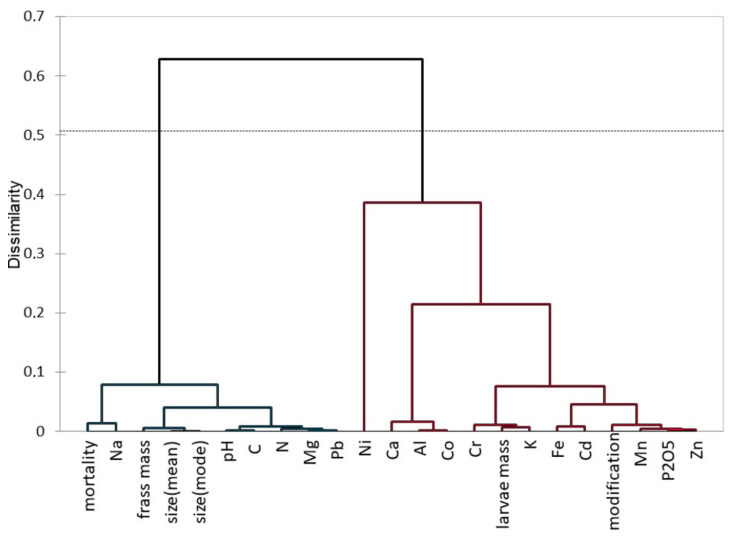
Dendogram of AHC showing dissimilarity between parameters.

**Table 1 materials-15-07516-t001:** Properties of EPS used.

Type of EPS	Apparent Density [kg/m^3^]	Volume of Used Specimen [cm^3^]
Ordinary—white	24.31 ± 0.50	160.0 ± 5
With addition of graphite—grey	19.66 ± 0.50	160.0 ± 5

**Table 2 materials-15-07516-t002:** Larvae population and frass characteristics.

Observation	Unit	White EPS	Grey EPS
Larvae mass	[g]	292.89	327.81
Mortality	[g]	207.11	172.19
Frass mass	[g]	35.61	33.01
Mean size	[µm]	299.7	282.7
Mode size	[µm]	379.3	357.1
Standard deviation	[µm]	20.09	33.26

**Table 3 materials-15-07516-t003:** Chemical composition of frass [g/kg].

Compound	White EPS	Grey EPS
P_2_O_5_	0.655	0.791
N	105.900	106.500
C	378.800	371.000
Ca	15.643	23.327
Mg	2.981	2.9644
K	16.177	17.618
Na	1.365	1.198
Fe	0.547	0.739
Al	0.922	1.443
Mn	0.046	0.056
B	0.000	0.000
Co	0.000	0.000
Ni	0.000	0.002
Cu	0.000	0.000
Zn	0.088	0.105
Cr	0.002	0.002
Cd	0.000	0.000
Pb	0.006	0.006

## Data Availability

Upon request, data is available from corresponding author.

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
