# Peer review of "Concept of Sustainable Demolition Process for Brickwork Buildings with Expanded Polystyrene Foam Insulation Using Mealworms of *Tenebrio molitor"

_materials, 2022, doi:10.3390/ma15217516_

Round 1

Reviewer 1 Report

This study aims to prove the concept using two models of three-layered brickwork wall which is very common in Europe. The proposed approached was successful. Its possibilities and limitations were discussed and analysed. By using mealworms of Tenebrio molitor one can make the demolition process much cleaner.

The article is well research and contains novel idea that adds some information to the body of knowledge. Likewise, the paper complies with the writing standard of the Journal and all tests were done according to the normal standard of tests. Based on these aforementioned, I recommend that the research paper can be accepted for publication after the minor revisions is corrected according to suggestions.

1. Please use new Roman font for the full text, and add line numbers for easy review.

2. The abstract should include four parts: purpose, method, result and conclusion. The author should refine them again.

3. Remove the space between is and a in the first row in

4. The conclusion is too simple. Please refine it further.

5. How efficient does the author use this biological method?

6. Whether new harmful or toxic organic residues will be generated when EPS is treated by biodegradation process, the author needs to further explain.

Author Response

Reviewer #1

  1. Please use new Roman font for the full text, and add line numbers for easy review.

The manuscript was prepared according requirements of MDPI. Palatino Linotype is the font used by MATERIALS. Authors have no other option as far as font is concerned.

Line numbers were added.

  1. The abstract should include four parts: purpose, method, result and conclusion. The author should refine them again.

The abstract was amended as suggested.

  1. Remove the space between is and a in the first row in

The manuscript was prepared according to requirements of MDPI. All typesetting mistakes were amended.

  1. The conclusion is too simple. Please refine it further.

Conclusions were extended. Taking into account the proof of concept character of the research programme only key conclusions could be drawn. More refine conclusions will be possible to draw after the execution of the subsequent research programme based on much larger specimens and populations of results.  

  1. How efficient does the author use this biological method?

The tests were conducted only in a lab scale. After successfully proving the concept the tests on much larger scale are being prepared (including testing of additional properties and characteristics). 

  1. Whether new harmful or toxic organic residues will be generated when EPS is treated by biodegradation process, the author needs to further explain.

The comment was added making this matter clear.

Reviewer 2 Report

-          Whole the manuscript should be completely checked with native expert.

-          The main achievements of the project should be brought at the end of the Abstract section.

-          All of the references should be write in same style:

               For example ref No. 2, the volume and start and end page were written but, in Refs 1,3 and 4            none of the mentioned parameters were not written.

-          The literature review is not well:

        There are just four 2021 ref and NO 2022 ref!

-          The Properties of used EPS in Table 1 should be reported in average (+-) standard deviation. (at least with 3 times repeat)

-          Drawn figures should be re-drawn with average (+-) standard deviation. In other words, the experiments should be done at least for 3 times and data should be analyzed with standard deviation. Moreover, statistically significant differences should be calculated and reported.

Author Response

Reviewer #2

  • Whole the manuscript should be completely checked with native expert.

The paper went through the linguistic proof-reading offered by MDPI.

  • The main achievements of the project should be brought at the end of the Abstract section.

The Abstract was amended. Suggested comments were added.

  • All of the references should be write in same style. For example ref No. 2, the volume and start and end page were written but, in Refs 1,3 and 4 none of the mentioned parameters were not written.

The style of all references was homogenized according to requirements of MDPI.

  • The literature review is not well: There are just four 2021 ref and NO 2022 ref!

Some scientific papers from 2022 were added to the reference list (and cited).

The literature references were used according to their relevance to the topic of the paper. Authors do not have any influence on EU directives and other official documents which are valid but at the same time can be quite dated.

Considering only 24 references to scientific papers currently there are: 3 from 2022, 2 from 2021, 4 from 2020, 3 from 2019, 8 from 2018 and 4 older.

  • The Properties of used EPS in Table 1 should be reported in average (+-) standard deviation. (at least with 3 times repeat).

It was amended.

  • Drawn figures should be re-drawn with average (+-) standard deviation. In other words, the experiments should be done at least for 3 times and data should be analyzed with standard deviation. Moreover, statistically significant differences should be calculated and reported.

Information about deviation was added to Table 1. Both types of used EPS were tested using multiple specimens. The bio-processing of EPS was proved only on one set of specimens, thus showing statistical differences of properties of the achieved frass would be pointless. A new research programme is being prepared. It will be conducted on much larger specimens and using a statistically significant number of specimens (for each type of specimens). In this case full statistical analysis will be enabled.    

Round 2

Reviewer 2 Report

The manuscript is improved.